# Reliability of Different Bending Test Methods for Dental Press Ceramics

**DOI:** 10.3390/ma13225162

**Published:** 2020-11-16

**Authors:** Daisuke Miura, Yoshiki Ishida, Taira Miyasaka, Harumi Aoki, Akikazu Shinya

**Affiliations:** 1Department of Dental Materials Science, School of Life Dentistry at Tokyo, The Nippon Dental University, 1-9-20, Fujimi, Chiyoda-ku, Tokyo 102-0071, Japan; daisuke@tky.ndu.ac.jp (D.M.); yishida@tky.ndu.ac.jp (Y.I.); miyasaka@tky.ndu.ac.jp (T.M.); haruaoki@tky.ndu.ac.jp (H.A.); 2Department of Life Dentistry, School of Life Dentistry at Tokyo, The Nippon Dental University, 1-9-20, Fujimi, Chiyoda-ku, Tokyo 102-8159, Japan; 3Department of Prosthetic Dentistry and Biomaterials Science, Institute of Dentistry, University of Turku, Lemminkaisenkatu 2, 20520 Turku, Finland

**Keywords:** dental press molded ceramics (DPCs), weibull coefficient, biaxial flexural test

## Abstract

Objective: This study investigates the reliability of different flexural tests such as three-point-bending, four-point bending, and biaxial tests, in strengthening the dental pressed ceramics (DPCs) frequently used in clinical applications. Methods: The correlations between the three types of bending tests for DPCs were investigated. Plate-shaped specimens for the three-point and four-point bending tests and a disc-shaped specimen for the biaxial bending test were prepared. Each bending test was conducted using a universal testing machine. Results: The results for six DPCs showed that the flexural strength in descending order were the three-point flexural strength, biaxial flexural strength, and four-point flexural strength, respectively. Then, a regression analysis showed a strong correlation between each of the three test methods, with the combination of four-point and biaxial flexural strength showing the highest values. The biaxial flexural strength was not significantly different in the Weibull coefficient (m) compared to the other tests, with the narrowest range considering the 95% interval. The biaxial bending test was found to be suitable for materials with small plastic deformation from the yield point to the breaking point, such as DPCs.

## 1. Introduction

In recent years, ceramics have been widely used to fabricate crown restorations due to the increasing demand for aesthetic properties, mechanical strength, and biocompatibility [1,2]. All-ceramic crowns are fabricated via the refractory model, casting, computer-aided design/computer-aided manufacturing (CAD/CAM), and press mould techniques. In the refractory model technique, crowns were fabricated via the building and baking method on the refractory casts made from duplicate impressions. However, the technique required expert skills and a long working time [3]. The casting technique has high strength and good optical properties owing to the crystallizing process after casting. However, this technique has some disadvantages such as crystallization inhomogeneity, casting porosity, and producing a rounded casting instead of metal casting due to the high sensitivity of ceramics to the temperature of the casting mould [4].

The CAD/CAM technique has various advantages such as high precision, equal quality, and less crown fabrication time. However, it is not easy to mill a complex shape [5]. In the press mould technique, glass-ceramic ingots are heated from 1100 to 1200 °C for a steady flow, and then pressurized injection is injected into the casting mould made via the lost-wax process. Since calipers are absent from the crystallization process, the press mould technique can easily produce a crown, it is more stable, has greater strength, as well as better adaptability when compared to the casting technique. Moreover, it is possible to reproduce a colour tone close to that of natural teeth using the layering method [6]. Since the 1980s, various improved techniques for the fabrication of all-ceramic restorations have been developed [7]. Previously, although the porcelain jacket crown had excellent aesthetic properties, it was not reliable in terms of strength [8]. Therefore, to improve the mechanical strength without reducing the optical advantage, various all-ceramic systems have been developed. In the dental pressed ceramics (DPCs), ingots were heated and pressed into casting models prepared via the lost-wax process with progressing crystallization. Therefore, the DPCs are easy to use since they do not require crystallization and have great mechanical properties. DPCs show superior advantages to other materials and they are frequently used in clinical applications. This is due to the fact that there is less technical error and the crown shape can be accurately reproduced [6,9]. In particular, the Li_2_O_5_Si_2_ -based DPCs show greater mechanical properties compared to other DPCs. Therefore, it is useful not only for crown restorations but also for inlay, onlay, and dental implant superstructures [10,11,12,13].

It has been proposed that the three-point, four-point, and biaxial flexural tests are performed for evaluating the flexural strength of dental ceramic materials by the International Organization for Standardization (ISO) [14]. However, there are some factors such as making methods of the specimens, test protocols, or strength of the materials to be considered when measuring the flexural strength. Moreover, the stress distribution in the specimens might affect the strength due to the brittleness of ceramic materials. It is known that preventing the edge effect influence from the shape on rectangular specimens is too difficult. Therefore, to measure the strength of dental ceramics, biaxial flexural tests are used [15,16]. Biaxial flexural tests provide reliable values, since the maximum stress is located in the centre of the specimen and the effect of burrs and cracks on the edges can be reduced [17]. There is a report that the biaxial flexural test method is more reliable compared to the three-point and four-point bending tests for the ceramics, since the Weibull modulus in the biaxial flexural test was larger than that of the other bending test methods [18]. In addition, the probability of failure can be evaluated using the Weibull distribution function based on the weakest ring theory. There are some reports on the strength of the DPCs using one or two types of bending tests [17,19]. We have reported that there are correlations between two types of bending tests. However, there are few reports comparing the three test methods for DPCs strength. In this study, we conducted a series of experiments to clarify the correlation between three types of bending tests on the strength of DPCs.

## 2. Materials and Methods

The materials used are listed in Table 1. To make specimens, four different dental ceramics made from leucite such as vintage ZR (VI; Shofu, Kyoto, Japan), VITA PM9 (PM; VITA, bad Säckingen, Germany), Cerabien ZR (CE; Kuraray Noritake Dental, Tokyo, Japan), and GC Initial PC (PC; GC, Tokyo, Japan), and two types of Li2O5Si2-based DPC, i.e., SI-C1301 (SI; Shofu) and IPS e-max (EM; Ivoclar Vivadent, Schaan, Liechtenstein) were used. The preparation of the specimens and the test measurements of flexural strength were done according to the ISO (6872) standard for ceramics [14].

Using a low speed cutting machine (Isomet 1000 precision saw, BUEHLER, Lake Bluff, IL, USA), the acrylic plate of the three-point and four-point bending tests was cut to plate-shaped specimens (1.2 mm height × 4.0 mm width × 25 mm long). For the biaxial flexural test, the acrylic plate was cut to disc-shaped specimens (16 mm diameter × 1.2 mm thickness). Thereafter, the specimens, two plates, and a disc-shaped acrylic pattern were placed on a crucible former at a 45°-angle using sprue waxes (Ready Casting WAX, GC) to occupy 16.0 mm of the total height. The investment (Press VEST, Ivoclar Vivadent) was mixed with 60% distilled water in a vacuum mixing machine for 150 s. Subsequently, the patterns were embedded. The VI, PM, and PC were press-formed by the Multimat NT press (Dentsply Sirona, York, PA, USA) after pre-heating for 1 h, SI was press-formed by the Esthemat press (Shofu), and the CE and EM were press-formed by the Programmat EP 600 (Ivoclar Vivadent). The press forming schedule of each product was based on the manufacturer’s recommendations (Table 2).

Test specimens for the three-point and four-point bending tests were formed into plate shapes (1.2 mm height × 4.0 mm width × 25 mm long) using sic paper (#1000). Then, the exact values of the samples were checked with a caliper (Shinwasokutei, Saitama, Japan). Thereafter, they were polished with alumina powder (1.0 μm). The edges of the plate-shaped specimens were chamfered to be 0.07 mm at 45°. All specimens were stored in distilled water at 37 ± 2 °C for 24 ± 2 h. A universal testing machine (AGS-X, Shimadzu, Kyoto, Japan) was used for carrying out all bending tests with a crosshead speed of 1.0 mm/min at room temperature. A schematic diagram of the specimens and instruments used in each bending test is shown Figure 1.

The three-point bending test was performed in accordance with ISO [14]. The flexural strength σ (MPa) is represented by formula (1), where P is the load at failure, L is the distance between the supports, which is equal to 20 mm, and w and t are the width and thickness of the specimen, respectively, measured prior to testing:σ = 3PL/2wt^2^(1)

The four-point bending test was performed in accordance with ISO [14]. The flexural strength σ (MPa) is represented by formula (2), where P is the load at failure, L is the distance between the supports, which is equal to 20 mm, L1 is the load span (center-to-center distance between inner loading rollers), and w and t are the width and thickness of the specimen, respectively, measured prior to testing:σ = 3P × (L − L_1_)/2wt^2^(2)

The biaxial flexural test performed by means of the fixture and support of the test specimen was three steel balls with a diameter of 3.0 mm and was positioned 120° apart on a support circle with a diameter of 10 mm. The specimens were placed on the supporting balls so that the load was applied at the center. Then, a 50 µm polyethylene film was placed between the supporting balls and the specimen, and the loading piston and specimen were placed. The load was applied with a flat plunger with a diameter of 1.4 mm at the center of the specimen. The biaxial flexural strength σ (MPa) was calculated from formula (3), where P (N) is the load at failure and t (mm) is the thickness of the specimen.
σ = −0.2387 × P × (X − Y)/t^2^(3)
where X and Y are given as follows (4):X = (1 + ν) ln(r_2_/r_3_)^2^ + [(1 − ν)/2](r_2_/r_3_)^2^
Y = (1 + ν) [1 + ln(r_1_/r_3_)^2^] + (1 − ν)(r_1_/r_3_)^2^(4)
in which ν is Poisson’s ratio (assuming 0.25), r_1_ (mm) is the radius of support circle, r_2_ (mm) is the radius of loaded area, r_3_ (mm) is the radius of specimen.

The number of repetitions for all bending tests was set at 20. Two-way ANOVA (factor A as the type of material, factor B as the test method) and Tukey’s multiple comparisons were performed using the flexural strength obtained. Weibull analysis was also performed for the results, and the parameters were calculated based on the maximum likelihood method (JMP 11, SAS Institute Japan, Tokyo, Japan).

## 3. Results

The average flexural strength is shown in Figure 2. The SI and EM of Li_2_O_5_Si_2_ showed a higher flexural strength in the three-point bending test, whereas the four-point bending test showed the smallest flexural strength among all the test methods in all materials. The order of the mean value in all materials was the three-point bending test ≥ biaxial flexural test > four-point bending test, with no significant differences between the test methods except for some products. Significant differences (*p* < 0.01) were observed in all materials when comparing the three-point and four-point flexural strength. In comparison with the four-point and biaxial flexural strength, a significant difference (*p* < 0.01) was observed in PC, SI, and EM. However, no significant difference was observed in the VI, PM, and CE groups (*p* > 0.05). There was no significant difference in the VI, PM, and CE (*p* > 0.05). Figure 3 shows the correlations among the test methods using the flexural strength obtained. High determination coefficients were found in all combinations of test methods. In particular, the highest determination coefficient was observed between the three-point and four-point flexural strength.

The Weibull and determination coefficients obtained by Weibull analysis using the flexural strength of each test are shown in Table 3. Capital letters indicate values with significant differences for the Weibull coefficient (m) with 95% confidence intervals, whereas lowercase letters indicate values with no significant difference for the m with 95% confidence intervals. In the biaxial flexural strength, there was no significant difference in the m compared with other tests, and the range of the considering 95% interval was the narrowest. Weibull plots obtained by Weibull analysis using the flexural strength of each test method are shown in Figure 4. Figure 5 shows the correlations between the Weibull characteristic strength (S_0_) of the three test methods. There was a significant difference when the three test methods were combined, and a large determination coefficient was observed for each combination. In particular, the S_0_ of the biaxial and four-point flexural strength showed large values.

## 4. Discussion

The measured values of the three flexural strengths obtained in this study are ranked in the following order: Three-point flexural strength ≥ biaxial flexural strength > four-point flexural strength. The Li_2_O_5_Si_2_-based SI and EM exhibited a larger flexural strength compared to other ceramics. This is due to the fact that Li_2_O_5_Si_2_ ceramics are high-density porous materials when compared with KAlSi_2_O_6_ ceramics [8]. During the comparison of the intensities of all six DPCs used in this study, there was no significant difference in the combination of DPCs with nearly equal intensities such as VI, PM, CE, and PC, where the crystals were KAlSi_2_O_6_. Significant differences were observed between the Li_2_O_5_Si_2_-based SI and EM for the three-point and four-point bending tests. However, no significant difference was observed in the biaxial flexural test. A comparison of the materials revealed significant differences between the three-point and the four-point bending tests in the VI, PM, and CE. In the PC and SI, there were significant differences between the three-point and four-point bending tests and the four-point bending test and biaxial flexural test. In the EM, there were significant differences among all bending tests. Bending tests are based on the theory of tensile stresses in the lower side of the specimens. For brittle materials such as ceramics, the strength of the material is not only dependent on the maximum stress, but also the stress distribution in the specimen. It is assumed that there was a significant difference among the test methods, since Li_2_O_5_Si_2_ is denser and stronger than KAlSi_2_O_6_, and the stress distribution of the specimens might be uniform [20]. From these results, it is possible to rank the Li_2_O_5_Si_2_-based DPCs that would be clinically useful accurately. As shown by the results of the regression analysis using the flexural strength average obtained from each bending test (Figure 3), the measurement coefficients of the biaxial and the four-point flexural strength were the largest. In addition, there were small differences from regression lines. Generally, the bending moment of the four-point bending test becomes constant between the load points. Additionally, the moment in the biaxial flexural test becomes constant between the edges of the pistons. Moreover, the biaxial flexural test can be considered as the four-point bending test expanded in two dimensions. Therefore, the highest coefficient of determination of the regression was found between the biaxial and four-point flexural strength (R^2^ = 0.989).

The measurements for each test were dependent on the size of the sample and decrease with an increase in volume. In particular, the measured values of the biaxial bending sample are smaller compared to those of the three-point bending sample. This indicates that the strength of ceramics is directly affected by the size and distribution of defects, which are the starting points of failure. However, the Weibull analysis results indicate that the m ranges from 3.84 to 16.3, indicating no particular change in the failure mode [21,22]. The Weibull characteristic strength (S_0_), which is the strength occurring at a failure probability of 63.2%, showed the same trend as the average flexural strength, but was larger than the average value (Table 3). There was no significant difference in the m of the biaxial flexural strength of the DPCs, and the 95% confidence interval was the narrowest. Although the values were slightly different for each test method and specimen shape, there was no significant difference between the methods in the m. Therefore, the m, which indicates the magnitude of the change in intensity, might not depend on the specimen volume. Moreover, it was found that the biaxial bending test was as reliable as the other tests in DPCs. The results of the regression analysis using S_0_ for the combination of the four-point and biaxial flexural strength showed the highest values.

Despite the tensile test being the most sensitive to defects as a strength test of the materials, there are some reports that the bending test is more sensitive to the strength evaluation of brittle materials such as ceramics [1,9]. In addition, the four-point bending test is preferable for evaluating the strength in terms of an effective cross-sectional area, since the tensile stress is distributed to the entire lower surface of the specimen in the three-point bending test. Biaxial and four-point bending tests are suitable for evaluating the strength of brittle materials, such as ceramics due to the fact that the stresses applied by the piston are uniformly distributed over the entire specimen. Therefore, the plastic deformation from the yield point to the fracture point is small [23,24]. Our results showed that the biaxial flexural test shall be a potentially effective method in various bending tests to measure the flexural strength on DPCs, as reliable as the four-point bending test.

## 5. Conclusions

In this study, the reliability of various bending tests, such as three-point bending, four-point bending, and biaxial flexural tests, were investigated to improve the strength of DPCs, which are often used in clinical practice. A three-point, four-point, and biaxial flexural tests were conducted for six dental press-formed ceramics to investigate a reliable bending test method. In addition, we also analyzed the correlation among the test methods. The biaxial flexural strength was not significantly different in m compared to the other tests, with the narrowest range considering the 95% interval. Then, a regression analysis using S_0_ showed a strong correlation between each of the three test methods, with the combination of four-point and biaxial flexural strength showing the highest values. The biaxial flexural test was found to be suitable for materials with small plastic deformation from the yield point to the breaking point, such as DPCs.

## Figures and Tables

**Figure 1 materials-13-05162-f001:**
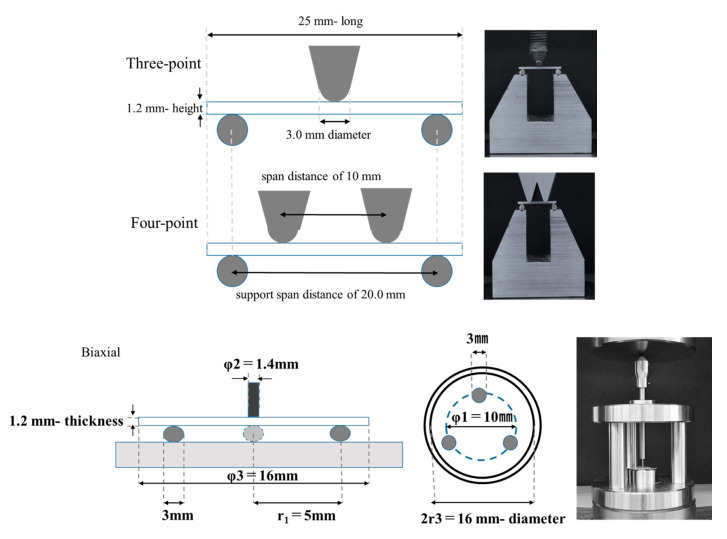
Schematic diagram of the equipment used in each bending test. Upper: Three-point bending test, Middle: Four-point bending test, Lower: Biaxial bending test.

**Figure 2 materials-13-05162-f002:**
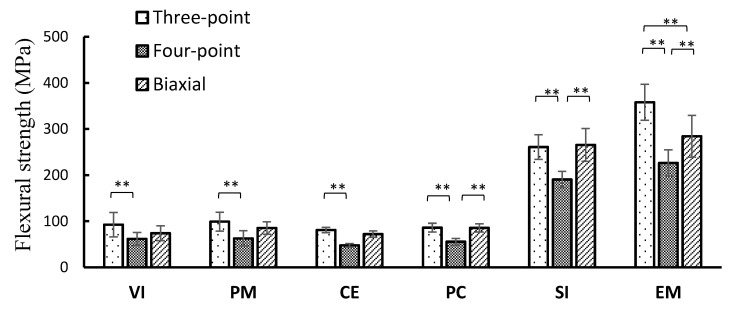
Average flexural strength (** *p* < 0.05, Standard deviation).

**Figure 3 materials-13-05162-f003:**
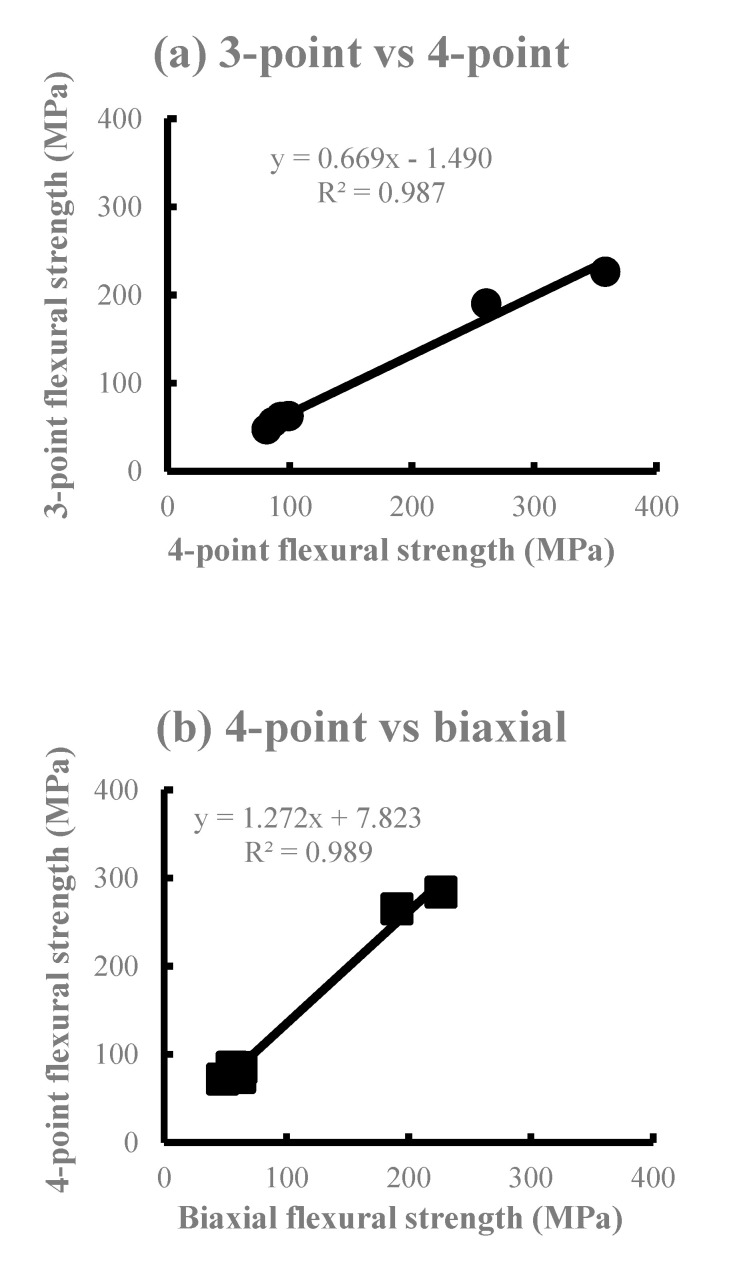
The correlations between the average flexural strength of the three test methods. (**a**) 3-Point vs. 4-point, (**b**) 4-point vs. biaxial, (**c**) 3-point vs. biaxial.

**Figure 4 materials-13-05162-f004:**
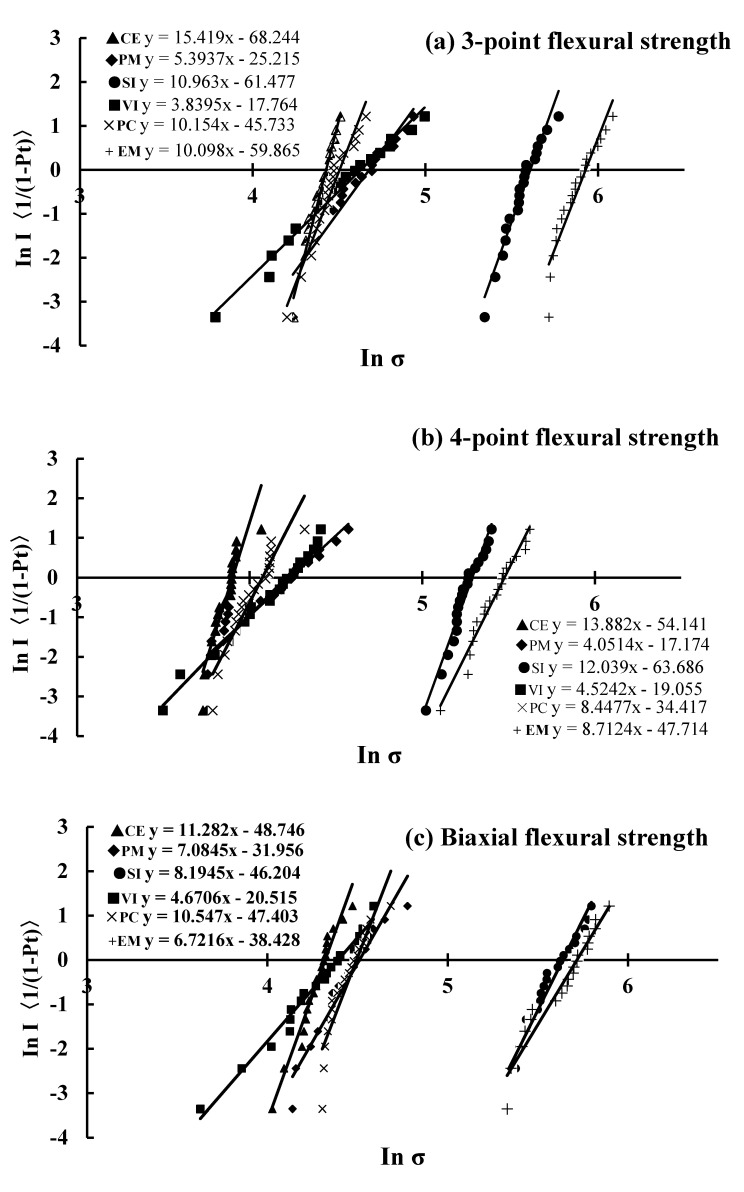
Weibull plots for each test. (**a**) 3-Point flexural strength, (**b**) 4-point flexural strength, (**c**) biaxial flexural strength.

**Figure 5 materials-13-05162-f005:**
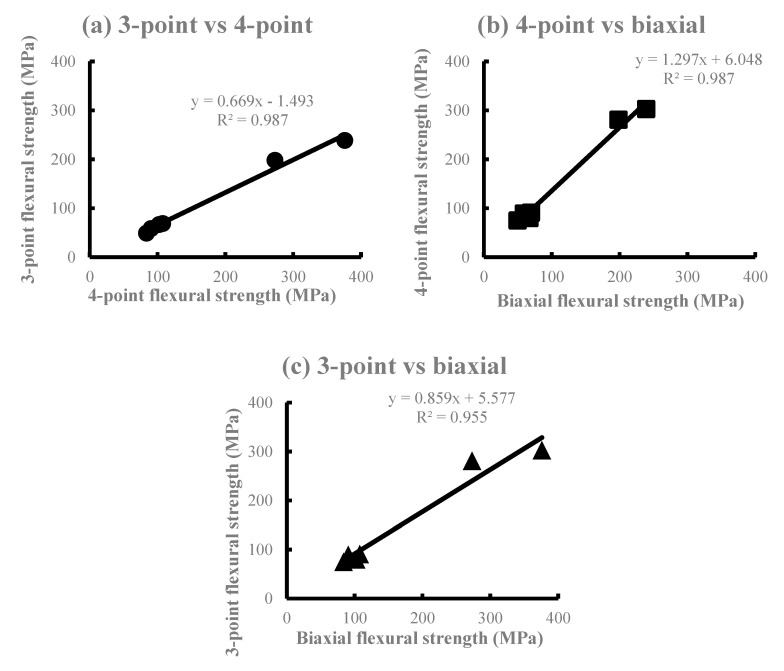
The correlations between the Weibull characteristic strength (S_0_) of the three test methods. (**a**) 3-Point vs. 4-point, (**b**) 4-point vs. biaxial, (**c**) 3-point vs. biaxial.

**Table 1 materials-13-05162-t001:** DPCs used in this study.

Product Name	Manufacturer	Code	Main Crystal
Vintage ZR	Shofu	VI	Leucite
VITA PM9	VITA	PM	Leucite
Cerabien ZR	Kuraray Noritake	CE	Leucite
GC Initial PC	GC	PC	Leucite
SI-C1301	Shofu	SI	Lithium disilicate
IPS e-max	Ivoclar Vivadent	EM	Lithium disilicate

**Table 2 materials-13-05162-t002:** Press forming schedule.

Material	Mold Size (g)	Starting Temperature (°C)	Heating Rate (°C/min)	Firing Temperature (°C)	Mooring Time (min)
VI	100	700	50	940	20
PM	100	700	60	960	20
CE	100	700	60	1045	15
PM	100	700	60	960	20
SI	100	700	60	910	15
EM	100	700	60	915	15

**Table 3 materials-13-05162-t003:** The Weibull and determination coefficients obtained by Weibull analysis using the flexural strength of each test.

Code	Weibull Coefficient (m)	Statistical Significance	95% Confidence Limits	Weibull Characteristic Strength (S_0_)	Statistical Significance	95% Confidence Limits
VI	3.84	A B C D	2.88–5.77	102.23	a b	90.60–115.34
PM	5.29	E	3.96–7.94	107.36	a	98.30–117.22
CE	16.30	A E	12.11–24.92	83.56		81.22–85.96
PC	9.71	B	7.32–14.44	90.43	b	86.21–94.86
SI	10.20	C	7.68–14.93	272.91		260.69–285.69
EM	9.56	D	7.21–14.21	375.89		358.06–394.61
VI	5.25	F G	3.87–8.26	67.09	c d	61.45–73.24
PM	4.12	H I J	3.09–6.22	69.16	c e	61.80–77.39
CE	11.60	F H	8.95–16.46	49.44		47.49–51.46
PC	7.50		5.73–10.85	58.87	b e	55.32–62.63
SI	11.91	G I	8.93–17.85	198.34		190.76–206.23
EM	8.83		6.60–13.32	238.98		226.76–251.85
VI	5.43		4.01–8.42	80.30	f g h	73.78–87.41
PM	6.48		4.92–9.47	91.14	f i	84.83–97.92
CE	10.96		8.30–16.10	75.19	g	72.08–78.44
PC	9.37		7.15–13.59	89.59	h i	85.25–94.15
SI	8.05		6.02–12.13	281.30		265.54–298.00
EM	7.46		5.52–11.48	303.18		284.99–322.53

Identical uppercase letters indicate that the values are statistically different (*p* < 0.05). Identical lowercase letters indicate that the values are not statistically different (*p* > 0.05).

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
