# Peer review of "Reliability of Different Bending Test Methods for Dental Press Ceramics"

_materials, 2020, doi:10.3390/ma13225162_

Round 1
Reviewer 1 Report
You can find attached my comments.

Author Response
Comments for reviewer 1
Dear Authors
This study aims to evaluate reliability of different bending test methods (three-point-bending, four-point bending, and biaxial flexural tests) for dental press ceramics. The correlations between the three types bending tests were investigated.
Thank you very much for the valuable suggestions. I have corrected all the points based on reviewer’s advice.
GENERAL
- A non-breakable space must be used between units and values.
A space has been added between units and values in the text.
- In the text, reference numbers should be placed before the punctuation.
The reference numbers have been properly placed.
MATERIALS AND METHODS
- Table 1 layout must be revised.
The position in Table 1 has been corrected.
- “into plate shapes(1.2 mm- height × 4.0 mm-width × 25 mm-long ) using sic paper(♯1000).” Spaces must be added.
A space has been added between units and values in the text.
- How are the samples dimensions and geometry verified? This point must be added.
Added the following minutes to Materials and Methods.
“The exact values of the samples were then checked with a caliper.”
- To clarify more “m” in Table 3 header must be replaced by ”Weibull coefficient (m)”.
Table 3 Replacing the "m" in the header with the "Weibull coefficient (m)" in the header.
- To clarify more “S0” in Table 3 header must be replaced by ”Weibull characteristic strength (S0)”
“S0” in the header of Table 3 was replaced with the “Weibull characteristic strength (S0)”.
- Spaces must be added in figure 2 legend axis.
A space has been added to the legend axis in Fig 2.
- Figures 3 is too small. Figure 3 quality must be improved.
Fig 3 is divided into three graphs: "3-point vs 4-point", "4-point vs biaxial", and "3-point vs biaxial".
Accordingly, Fig 3 has become Fig 4.
Reviewer 2 Report
Authors have introduced a need to figure out the most effective method for flexural testing for Dental Pressed Ceramics (DCPs). They have mentioned about three point, four point and biaxial flexural test in this regard. They have presented the the results of all different methods and made a conclusion that biaxial flexural test is the best among all flexural mechanical test. The conclusion seems logical based on the results shown.
Manuscript can be stronger if authors compared their data with the data reported in the literature. Manuscript needs some fix in terms of language, some sentences are un-necessarily repeated.
Manuscript is acceptable with revisions.
Author Response
Thank you very much for the valuable suggestions. I have corrected all the points based on reviewer’s advice.
Reviewer 3 Report
It is recommended that authors consider modifying highlighted sections in the attached PDF file. Additionally, the following comments are provided for further clarification of these highlights and improvement of this work:
1) Certain details can be removed from the abstract section and explained in the Materials and Methods section.
2) The introduction section, which is currently made up of only one paragraph, can benefit from further in-depth review of the literature. The main function of the introduction section is to show a research gap or establish a novel and valuable hypothesis. It also should justify the decisions that authors have taking in the methodology section. For example, why certain procedures are implemented as they have, why certain materials are selected versus other ones, why certain statistical techniques are used, etc. Although authors have indicated the novelty of their work, the significance of their findings is not clarified. It is also not explained how this research can improve the current practices in science and/or industry.
3) The following sentence in the introduction section requires an appropriate citation: “It has been proposed that the three-point, four-point, and biaxial flexural tests are performed for evaluating the flexural strength of dental ceramic materials by the International Organization for Standardization (ISO).”
4) It is strongly recommended that authors provide pictures/schematics that can clarify the appearance and dimensions of the test setup as well as the specimens.
5) Table 1 is surrounded by the text from Materials and Methods section which reduces its legibility. This table must be separated from the body text and must be accompanied with an informative caption.
6) The writing of some chemical formula is inconsistent throughout the manuscript.
7) It is recommended that in the Materials and Methods section, authors better explain the equations that they have used for their statistical analysis.
8) In Materials and Methods section, the writing of some numbers and their units are incorrect or inconsistent.
9) In Fig. 1, only one of the columns is accompanied by the error bar. However, all columns must be presented with their corresponding error bars. Additionally, it must be clarified in the caption what the error bar represents (standard error, standard deviation, etc.)
10) In Fig. 2, 3, and 4, more attention must be paid to the significant digits of the information provided from the fits.
11) Fig. 2 and 4 in their current forms are very vague. It is not clarified which fitting equation and R-squared value belongs to which data set. Also, it is not clear which axis represents which test for each set of data points.
12) In all the figures and throughout the text, wherever an average value is reported, it must be accompanied by the corresponding standard error.
13) Fig. 3 must have subsections titled (a), (b) and (c). Then in the caption, each one must be explained separately.
14) The legends in Fig. 3 are scattered which makes it very crowded and difficult to use.
15) Fig. 3 contains many fits all of which are very close to one another. Therefore, it must be clarified which line belongs to which data set. Also, it is recommended to properly expand the horizontal axis while removing the sections that are not used.
16) In Fig. 4, the legend is incomplete.
17) In the results section, parameters “m” and “S0” are explained more than once.
18) The first paragraph of the Discussion section can be merged with introduction section. Such reviews do not belong to the results section.
19) The following inequality is presented in the results section: “three-point bending strength>biaxial flexural strength>four-point bending strength”. This is inconsistent with the same inequality presented in the Materials and Methods section as: “three-point bending test ≥ biaxial flexural test > four-point bending test”. Please provide the correct formulation and only once, preferably in the results section.
20) The following sentence in the results section requires an appropriate citation: “there are some reports that the bending test is more sensitive to the strength evaluation of brittle materials such as ceramics”.
21) In the conclusion section the following sentence is provided: “The biaxial bending test method is more effective for dental press moulded ceramics compared to the other two test methods”. However, in this section authors must better explain how they have arrived at this conclusion.
22) The conclusion section is concise, but not very informative. The conclusion section must reiterate the main goals of the paper, highlights of the methodology, the most important findings of the research, as well as any limitations.
23) Authors could consider evaluating and characterizing the quality of the samples that they have prepared prior to testing through CT or porosity measurement or any other method of their preference.
24) It is important that authors additionally study the results using specific strength parameter and explain any new conclusions.

Author Response
Comments for reviewer 2
Dear Authors
Thank you very much for the valuable suggestions. I have corrected all the points based on reviewer’s advice.
1) Certain details can be removed from the abstract section and explained in the Materials and Methods section.
Thank you very much for the suggestions. Revisions were made accordingly. Detailed information such as "machine (AGS-X, Shimadzu, Kyoto, Japan) was removed from Abstract.
2) The introduction section, which is currently made up of only one paragraph, can benefit from further in-depth review of the literature. The main function of the introduction section is to show a research gap or establish a novel and valuable hypothesis. It also should justify the decisions that authors have taking in the methodology section. For example, why certain procedures are implemented as they have, why certain materials are selected versus other ones, why certain statistical techniques are used, etc. Although authors have indicated the novelty of their work, the significance of their findings is not clarified. It is also not explained how this research can improve the current practices in science and/or industry.
The structure of the preface has been modified to clarify the reason for conducting this experiment. Reasons for the selection of the materials in this experiment and for the bending tests have been added to the preface.
"Since the 1980s, various improved techniques for the fabrication of all-ceramic restorations have been developed [7]. Previously, although the porcelain jacket crown had excellent aesthetic properties, it was not reliable in terms of strength [8]. Therefore, to improve the mechanical strength without reducing the optical advantage, various all-ceramic systems have been developed. In the Dental pressed ceramics (DPCs), ingots were heated and pressed into casting models prepared via the lost-wax process with progressing crystallization. Thus, the DPCs is easy to use because it does not require crystallization and has great mechanical properties. DPCs show superior advantages to other materials and they are frequently used in clinical applications."
3) The following sentence in the introduction section requires an appropriate citation: “It has been proposed that the three-point, four-point, and biaxial flexural tests are performed for evaluating the flexural strength of dental ceramic materials by the International Organization for Standardization (ISO).”
I have added references. Its reference number is 14 “ISO 6872-2008 Dentistry-Ceramic materials”.
4) It is strongly recommended that authors provide pictures/schematics that can clarify the appearance and dimensions of the test setup as well as the specimens.
Thank you for your variable advice and schematic test setup is added as a Fig. 1. Fig 1 shows a picture and diagram of the equipment used for the bending test.
5) Table 1 is surrounded by the text from Materials and Methods section which reduces its legibility. This table must be separated from the body text and must be accompanied with an informative caption.
The pointers have been corrected. Table 1 has been separated from the text.
6) The writing of some chemical formula is inconsistent throughout the manuscript.
The chemical formula is unified as Li2O5Si2. And it was changed.
7) It is recommended that in the Materials and Methods section, authors better explain the equations that they have used for their statistical analysis.
The formulas for the bending tests used in this experiment have been added to Materials and Methods. The formula for the three-point bend test is “σ = 3PL/2wt 2”, the formula for the four-point bend test is “σ = 3P (L-L1)/2wt 2”, and the formula for the biaxial bend test is “σ = -0.2387 P (X-Y)/t2”
8) In Materials and Methods section, the writing of some numbers and their units are incorrect or inconsistent.
Revised materials and methods. "4.0-mm wide × 25.0-mm long × 1.2 mm in height" has been corrected to "1.2 mm- height × 4.0 mm- width × 25 mm- long". And,"diameter:16mm x thickness 1.2mm" has been corrected to "16 mm- diameter × 1.2 mm- thickness".
9) In Fig. 1, only one of the columns is accompanied by the error bar. However, all columns must be presented with their corresponding error bars. Additionally, it must be clarified in the caption what the error bar represents (standard error, standard deviation, etc.)
I have added the standard deviation to the figure and included an explanation in the Fig 1.
10) In Fig. 2, 3, and 4, more attention must be paid to the significant digits of the information provided from the fits.
I have unified the number of significant digits in Figures 2, 3, and 4 to 3.
11) Fig. 2 and 4 in their current forms are very vague. It is not clarified which fitting equation and R-squared value belongs to which data set. Also, it is not clear which axis represents which test for each set of data points.
I have divided Fig 2 and Fig 4 into three graphs. "3-point vs 4-point", "4-point vs biaxial", and "3-point vs biaxial".
12) In all the figures and throughout the text, wherever an average value is reported, it must be accompanied by the corresponding standard error.
I have added the standard deviation to Fig 2.
13) Fig. 3 must have subsections titled (a), (b) and (c). Then in the caption, each one must be explained separately.
I have added (a), (b), and (c) to fig 3, and added an explanation to the caption.
14) The legends in Fig. 3 are scattered which makes it very crowded and difficult to use.
Fig3 have been corrected. I have rearranged the legends in Figure 3 to make them easier to see.
15) Fig. 3 contains many fits all of which are very close to one another. Therefore, it must be clarified which line belongs to which data set. Also, it is recommended to properly expand the horizontal axis while removing the sections that are not used.
In Fig 3, I have rearranged the fits. I have also made the horizontal axis consistent with a width of 3-6.5.
16) In Fig. 4, the legend is incomplete.
The legend in Fig 4 has been expanded and made appropriate.
17) In the results section, parameters “m” and “S0” are explained more than once.
Corrected the number of explanations for the parameters "m" and "S0" in the results section.
18) The first paragraph of the Discussion section can be merged with introduction section. Such reviews do not belong to the results section.
As you noted, the first paragraph of the discussion section has been revised and introduced in the introduction. The corrections are as follows.
“Since the 1980s, various improved techniques for the fabrication of all-ceramic restorations have been developed [7]. Previously, although the porcelain jacket crown had excellent aesthetic properties, it was not reliable in terms of strength [8]. Therefore, to improve the mechanical strength without reducing the optical advantage, various all-ceramic systems have been developed. In the Dental pressed ceramics (DPCs), ingots were heated and pressed into casting models prepared via the lost-wax process with progressing crystallisation. Thus, the DPCs is easy to use because it does not require crystallisation and has great mechanical properties. DPCs show superior advantages to other materials and they are frequently used in clinical applications. This is because there is less technical error and the crown shape can be accurately reproduced [6, 9]. In particular, the Li2O5Si2 -based DPCs show greater mechanical properties compared to other DPCs. Thus, it is useful not only for crown restorations but also for inlay, onlay, and dental implant superstructures [10-13]. It has been proposed that the three-point, four-point, and biaxial flexural tests are performed for evaluating the flexural strength of dental ceramic materials by the International Organization for Standardization (ISO) [14].”
19) The following inequality is presented in the results section: “three-point bending strength>biaxial flexural strength>four-point bending strength”. This is inconsistent with the same inequality presented in the Materials and Methods section as: “three-point bending test ≥ biaxial flexural test > four-point bending test”. Please provide the correct formulation and only once, preferably in the results section.
I have shown the inequality as "three-point bend test ≥ biaxial bend test > four-point bend test" in the Uniform Results column.
20) The following sentence in the results section requires an appropriate citation: “there are some reports that the bending test is more sensitive to the strength evaluation of brittle materials such as ceramics”.
I have added references. Its reference number is 19 “Effect of surface acid etching on the biaxial flexural strength of two hot-pressed glass ceramics”.
21) In the conclusion section the following sentence is provided: “The biaxial bending test method is more effective for dental press moulded ceramics compared to the other two test methods”. However, in this section authors must better explain how they have arrived at this conclusion.
(21), (22) ,(24). The conclusion section has been changed based on reviewers advice.
The conclusions were modified as follows:
“In this study, the reliability of various bending tests, such as three-point bending, four-point bending, and biaxial flexural tests, was investigated to improve the strength of DPCs, which are often used in clinical practice. A three-point, four-point, and biaxial flexural tests were conducted for six dental press-formed ceramics to investigate a reliable bending test method. In addition, we also analyzed the correlation amang the test methods. The biaxial flexural strength was not significantly different in m compared to the other tests, with the narrowest range considering the 95% interval. Regression analysis using S0 then showed a strong correlation between each of the three test methods, with the combination of four-point and biaxial bending strength showing the highest values. The biaxial flexural test was found to be suitable for materials with small plastic deformation from the yield point to the breaking point, such as DPCs, because the stress was distributed uniformly across the entire specimen and the bending strength was measured.”
22) The conclusion section is concise, but not very informative. The conclusion section must reiterate the main goals of the paper, highlights of the methodology, the most important findings of the research, as well as any limitations.
The conclusion has been changed based on reviewers advice. The main purpose and method highlights have been added. Please see and check new conclusion sited in reviewers comments in 21. .
23) Authors could consider evaluating and characterizing the quality of the samples that they have prepared prior to testing through CT or porosity measurement or any other method of their preference.
Thank you for the suggestion. We did not evaluate the quality of this experiment because we produced the samples according to the manufacturer's instructions. However, the air gap in the specimen may affect each bending test and will be clarified in future experiments.
24) It is important that authors additionally study the results using specific strength parameter and explain any new conclusions.
A comparison of the biaxial bending test with other test methods was added, based on the results obtained from the specific strength parameters. Please see and check new conclusion sited in reviewers comments in 21.
Round 2
Reviewer 3 Report
Thank you for all the improvements. However, there are still a few formatting issues that must be resolved. 1) All equations must be centered and numbered. 2) Body of the text must appropriately appear on the top and bottom of the figures and tables. Currently, the number of pages have unnecessarily increased because of poor formatting. For example, page 6 contains almost only 1 figure. Please follow the journal formatting instructions and look at the most recent publications from the journal for examples. Thank you,Author Response
Thank you very much for the suggestions. I have corrected the points you made.
1) All equations must be centered and numbered.
→The equations used in this case were centered and numbered in the Materials and Methods section. Please see in page 4.
2) Body of the text must appropriately appear on the top and bottom of the figures and tables. Currently, the number of pages have unnecessarily increased because of poor formatting. For example, page 6 contains almost only 1 figure. Please follow the journal formatting instructions and look at the most recent publications from the journal for examples.
→Thank you for your kind advice. The figures and tables used in this paper were arranged appropriately. Tables 1 and 2 were resized to fit on one page and placed on page 3.Also Figure 2 has been resized and the extra space has been removed. However in page 6 Table 3, I couldn’t manage it. I’m very sorry, and it well be modified by editing system before publication.